# Single Camera-Based Remote Physical Therapy: Verification on a Large Video Dataset

Jindrich Adolf [1,*], Jaromir Dolezal [1], Patrik Kutilek [2], Jan Hejda [2] and Lenka Lhotska[1,2]

1   Czech Institute of Informatics, Robotics and Cybernetics, Czech Technical University in Prague, Jugoslavskych partyzanu 1580/3, 16000 Prague, Czech Republic; jaromir.dolezal@cvut.cz (J.D.); lenka.lhotska@cvut.cz (L.L.)
2   Faculty of Biomedical Engineering, Czech Technical University in Prague, Sitna 3105, 272 01 Kladno, Czech Republic; kutilek@fbmi.cvut.cz (P.K.); jan.hejda@fbmi.cvut.cz (J.H.)
*   Correspondence: jindrich.adolf@cvut.cz

**Abstract:** In recent years, several systems have been developed to capture human motion in real-time using common RGB cameras. This approach has great potential to become widespread among the general public as it allows the remote evaluation of exercise at no additional cost. The concept of using these systems in rehabilitation in the home environment has been discussed, but no work has addressed the practical problem of detecting basic body parts under different sensing conditions on a large scale. In this study, we evaluate the ability of the OpenPose pose estimation algorithm to perform keypoint detection of anatomical landmarks under different conditions. We infer the quality of detection based on the keypoint confidence values reported by the OpenPose. We used more than two thousand unique exercises for the evaluation. We focus on the influence of the camera view and the influence of the position of the trainees, which are essential in terms of the use for home exercise. Our results show that the position of the trainee has the greatest effect, in the following increasing order of suitability across all camera views: lying position, position on the knees, sitting position, and standing position. On the other hand, the effect of the camera view was only marginal, showing that the side view is having slightly worse results. The results might also indicate that the quality of detection of lower body joints is lower across all conditions than the quality of detection of upper body joints. In this practical overview, we present the possibilities and limitations of current camera-based systems in telerehabilitation.

**Keywords:** remote therapy; markerless; pose detection; motion capture; physical rehabilitation; OpenPose; telerehabilitation; telemedicine

## 1. Introduction

The general concept of remote rehabilitation using motion capture (MoCap) systems has undergone a turbulent change in recent years, as there are several tools for three-dimensional assessments, including sophisticated automation technologies and algorithms, often costing time, expensive equipment and inapplicable inconvenience to the daily practice [1]. Telerehabilitation, or remote physical therapy, is one of the most common types of complex distance medicine that is applied in practice [2]. During recent years, a large number of MoCap systems detecting the pose of a human using a "markerless" approach have emerged [3], these systems work without the necessity of placement of any markers on the human body [4].

This approach reduces the technical and financial requirements and complexity of arrangement [5] and therefore it can be found in the context of distance medicine, not only in specialized clinics but also in the home environment [6].

Considering the application of distance medicine in the home environment, the most promising systems seem to be systems for the evaluation of body movements from commonly used standard video (RGB) records [7].

In this case, we only need a regular camera, which is currently integrated into most common electronic devices, such as smartphones or laptops, or smart TVs.

These systems have great potential for use mainly due to the reduced financial costs of purchasing these systems. These systems have reached such technical levels that they could be used in specific cases as alternatives to costly systems in clinics. However, these systems must also use special camera data processing software [8].

The most commonly used software tools for pose detection are OpenPose [9], Mask R-CNN [10], Google's Media Pipe [11], Alpha-Pose [12] all available as open source.

The time to the advent of markerless-based systems using neural networks is described in detail in Coyer's review [13]. All the systems mentioned above were operated only under laboratory conditions or used special HW.

All these software tools use artificial intelligence methods, namely neural networks (NN) trained on annotated images [14]. The datasets contain general static images of people in undefined positions, according to which NN learns to recognize anatomical points on the body. Existing benchmarks compare the speed and accuracy of detection using the above algorithms based on NN [14]. A shortcoming that limits wider use of low-cost MoCap systems and the mentioned software is the absence of evaluation of the validity of the data provided by these systems. This raises doubts about the use of telemedicine where it is necessary to know relatively accurately the information about the movement of specific anatomical points that physiotherapists need to monitor and modulate the rehabilitation intervention [15]. Thus, the main aim of our study is to determine whether the systems are sufficient to be used for home rehabilitation and under what specific conditions. We focus on the evaluation of the motion capture of different exercise positions and by different camera views, i.e., camera position relative to the subject. To achieve this aim, it is not possible to rely on existing benchmarks, but it is necessary to evaluate the efficiency of software use on video recordings of people moving while exercising. Thus, our study aims to validate the application of markerless systems using only one, generally positioned camera and thus applicable to home telerehabilitation.

*Related Work*

Studies [6,16] show how OpenPose and similar camera-based systems could be used in telerehabilitation, but they are not dealing with the practical telerehabilitation applicability. Studies that evaluate the accuracy of motion detection typically study only one specific type of exercise motion. Hernández [17] concludes that OpenPose is an adequate library for evaluating patient performance in rehabilitation programs that involve the following exercises: left elbow side flexion, shoulder abduction, and shoulder extension based on comparison with Kinect. Ota [8] verifies the reliability and validity of OpenPose-based motion analysis during the bilateral squat based on comparison with Vicon MoCap system. Studies [18,19] show that OpenPose can be used to capture and analyze both normal gait and abnormal gait. Nakano [20] compares the accuracy of 3D OpenPose with multiple cameras to the Vicon marker system. This study considers common human body movements such as walking, jumping, and ball throwing.

All the aforementioned studies use a system fixed in relation to the moving body and thus represent a one-sided task of interest, i.e., measuring body motion in only one anatomical plane. In general, when it comes to gait or run analysis, the setup is always the same, i.e., the camera is positioned to record only the sagittal plane of the body as accurately as possible. When it comes to measuring range of motion or measuring angles between joints, the setup is such that the person stands in a precisely defined position to the camera and only selected angles in a single anatomical plane are measured. Thus, none of the studies presented here considers the application of a camera based approach in distance rehabilitation and NN based software to recognize anatomical points, where the precise alignment of the camera system perpendicular to the anatomical plane being measured would not be necessary prior to measurement.

## 2. Methods

### 2.1. Design

The advantage of the camera-based approach is the ability to detect motion from any regular RGB video. This allows to use existing recordings and perform a large-scale evaluation. In this study, we choose to use the database created by PhysioTools. PhysioTools is one of the world's most comprehensive exercise libraries [21]. In our study, we consider only the ability of the system to estimate the human pose. We do not analyze the exercise itself. Therefore, we only extract the OpenPose reported confidence of the selected keypoints to infer the quality of detection. The design of our study is shown in Figure 1.

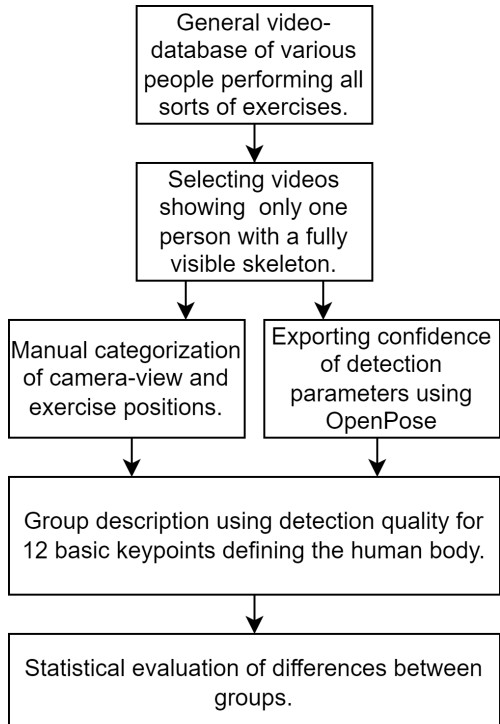

**Figure 1.** Study design—diagram explaining the sequence of steps.

For research purposes, we use a database based on an agreed template of commonly prescribed physical exercises printed from commonly used PhysioTools computer software (PhysioTools©, Product ID RG-PT1ENG, General Exercises Second Edition (English), Tampere, 339 Finland) [22]. PhysioTools is a database of professional trainers and serves as a video aid for exercises in the home environment. The average length of a video is 20 s. Videos have frame rates ranging from 25 fps to 50 fps, resolutions ranging from 0.1–0.6 Mpixel, and bit rate greater than 200 kbs. Our aim is to evaluate practical usability of recordings made in an uncontrolled home environment so we use no additional constraints on video quality.

Unlike typical studies [23] studying only one type of movement, the database we used is composed of hundreds of unique physical exercises, see Table 1 for quantity and Table 2 for categories.An exercise can be included in the database if it is performed by a single person and shows their whole body. Each video was manually checked to confirm that the entire trainee's body was in view throughout the recording. At the same time, manual categorization into specific groups was done by a single rater. Border cases were excluded from the analysis.

**Table 1.** Number of videos in each category. Subcategories are: lying down (LY); on the knees (Kn); sitting (Si); standing (St).

| Total (2133) | | | | | | | | | | | |
|---|---|---|---|---|---|---|---|---|---|---|---|
| Front View (357) | | | | ¾ View (1027) | | | | Side view (749) | | | |
| Ly | Kn | Si | St | Ly | Kn | Si | St | Ly | Kn | Si | St |
| 75 | 26 | 186 | 70 | 177 | 95 | 165 | 490 | 215 | 92 | 145 | 297 |

*2.2. Video Analysis*

Since the practical use assumes only one camera, we were interested in the influence of the orientation of the person towards the camera.

By analyzing the database, we determined the following three basic views to be the most common: the frontal view (frontal plane), the side view (sagittal plane), and the ¾ view, which is in between these planes, please see Figure 2. Although the ¾ plane is not biomechanically defined, it was most frequently used in instructional videos, because it provides an overview of the entire body and a better spatial understanding of records.

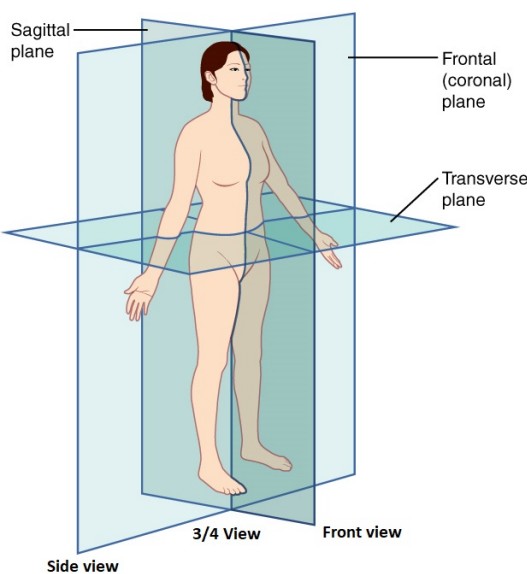

**Figure 2.** Orientation of the camera relative to the subject. Camera view options: front, ¾ and side [24].

**Table 2.** Categorization of videos.

| Starting Position | Camera View |
|---|---|
| lying down (Ly) | front view (frontal plane) |
| on the knees (Kn) | ¾ view |
| sitting (Si) | side view (sagittal plane) |
| standing (St) | |

*2.3. Keypoint Confidence Extraction*

The videos were analyzed using the OpenPose [25] algorithm. OpenPose uses a model with 25 keypoints. In the context of performing rehabilitation exercises, the body segments that are part of the appendicular skeleton are very often measured [26]. These segments allow for translational movement of the body through cyclic movements such as walking [27]. Thus, we used 12 points that allow us to determine the positions and

orientations of segments of the appendicular skeleton for further analysis. These 12 points are described in Table 3.

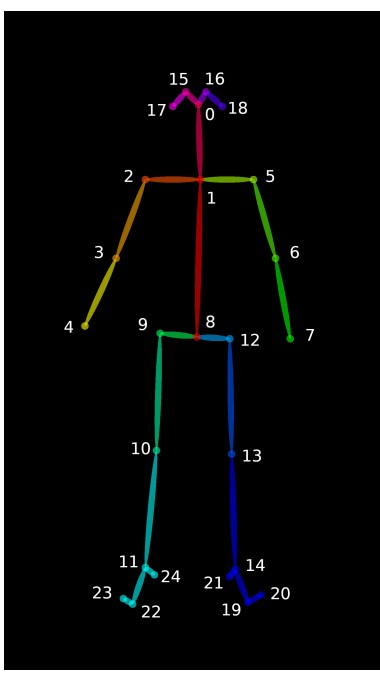

**Figure 3.** OpenPose 25 keypoints model [9].

**Table 3.** The 12 basic keypoints of the appendicular skeleton.

| No. | OP Name | The Most Appropriate Name of the Anatomical Point |
| --- | --- | --- |
| Upper body | | |
| 2 | RShoulder | R. acromion, end of the clavicle (collar bone)—top of shoulder |
| 3 | RElbow | R. lateral epicondyle of humerus, lateral epicondyle of the humerus. Outside of elbow. |
| 4 | RWrist | R. styloid process of the radius; wrist on thumb side. |
| 5 | LShoulder | R. acromion, end of the clavicle (collar bone)—top of shoulder |
| 6 | LElbow | R. lateral epicondyle of humerus, lateral epicondyle of the humerus. Outside of elbow. |
| 7 | LWrist | L. styloid process of the radius; wrist on thumb side. |
| Lower body | | |
| 9 | RHip | R. femur greater trochanter |
| 10 | RKnee | R. femur lateral epicondyle |
| 11 | RAnkle | R. fibula apex of lateral malleolus |
| 12 | LHip | L. femur greater trochante |
| 13 | LKnee | L. femur lateral epicondyle |
| 14 | LAnkle | L. fibula apex of lateral malleolus |

The outputs of OpenPose for each frame are the x,y coordinate values and the detection confidence for each of the 25 model points, please see Figure 3. From this information, we use only the confidence for the twelve points for each frame. This gives us 12 temporal signals for each unique video.

OpenPose processes each frame independently without using the time context. Repeated image analysis returns the same results. OpenPose has almost perfect test–retest reliability within device [8].

In our study, we are not dealing with the absolute position of the detected points. To evaluate the quality of detection, we use directly the confidence returned by OpenPose. Thus, we are not evaluating the accuracy of detection, but the detection capability itself.

Detection accuracy using annotated images is a classical metric for comparing machine learning algorithms; accuracy calculations are performed on large image datasets COCO [28], MPII [29]. In contrast, the evaluation of dynamic events has been studied only for single exercises and sub-joints [23]. Existing annotated 2D datasets deal with either images [30] or deal with a small variety of activities [31,32].

In contrast to the above-mentioned studies, we deal with many unique rehabilitation exercises.

OpenPose returns the confidence values of the keypoints in the interval of <0, 1>. Points that are not detected have a confidence value of zero. The confidence value is rarely used in single-camera setups because the position of the detected joints is accurate even with an average confidence value. On the other hand, in multi camera setup and 3D reconstruction tasks, which are very sensitive to misclassification, the confidence is used to weight joint positions [33] or to discard joints with a low confidence value, as the misclassification of points appears with values below 0.2 [34].

The result of the processed video is a matrix of 12 keypoint confidences in time.

### 2.4. Statistical Analysis

To compare the records, we calculated the medians of each time signal. This gave us 12 scalar keypoint confidence values defining each video. All records were assigned to exactly one of the subgroups, see Table 1. For each subcategory, we calculated the median confidence of the individual keypoints.

To visualize the results we chose box plots where outliers are not shown for clarity. We define outliers as elements more than 1.5 interquartile ranges above the upper quartile (75 percent) or below the lower quartile (25 percent).

The Kruskal–Wallis test was used to determine the statistical significance of group differences. Due to a large amount of data and the significant difference of one of the groups, all results were significant. Therefore, we decided to compare all groups, individually, with each other. To verify the normality of the data we used the Shapiro–Wilks test. The test has shown that values in the groups are not normally distributed. The groups also varied in size, see Table 1, therefore we used the Wilcoxon test to determine the statistical significance of the difference between the categories. All statistical calculations were performed using the Matlab (MATLAB and Statistics Toolbox Release 2019b, The MathWorks, Inc., Natick, MA, USA).

### 3. Results

In our study, we analyzed a total of 2133 videos. Each video shows only one trainee performing a unique exercise. Each video belonged to one of the "Camera View" and one of the "Starting position" groups. We present the results of our findings using OpenPose reported confidence values. We can use the confidence returned by the OpenPose algorithm as a measure of detection quality because it correlates with the percentage of correct keypoints metric [35].

The resulting median confidences for each joint and each category are shown in Table 4. Keypoints with a confidence value above 0.5 can be considered correctly detected [35]. These confidence values are associated with with clearly visible body parts [9].

Due to the large amount of data in the groups, the differences in confidence between groups are largely significant, although small in absolute terms. Therefore, for clarity, only differences that were not statistically significant (n.s.) are highlighted in the following boxplots. All the other differences were statistically significant, see Tables A1 and A2 in the Appendix A. Box-and-whisker plots of detection confidence for all the categories are show in Figures 4–9.

**Table 4.** Median confidence value for each specific group.

| Medians of Confidence for Selected Group and OP Keypoint | | | | | | | | | | | | |
| --- | --- | --- | --- | --- | --- | --- | --- | --- | --- | --- | --- | --- |
| Camera View | Front view | | | | ¾ view | | | | Side view | | | |
| Starting possition | Ly | Kn | Si | St | Ly | Kn | Si | St | Ly | Kn | Si | St |
| No.　　KP name | | | | | | | | | | | | |
| Upper body | | | | | | | | | | | | |
| 2　　RShoulder | 0.67 | 0.65 | 0.78 | 0.80 | 0.62 | 0.67 | 0.77 | 0.79 | 0.68 | 0.68 | 0.76 | 0.78 |
| 3　　RElbow | 0.75 | 0.81 | 0.82 | 0.84 | 0.76 | 0.81 | 0.82 | 0.84 | 0.78 | 0.82 | 0.81 | 0.83 |
| 4　　RWrist | 0.75 | 0.79 | 0.82 | 0.84 | 0.75 | 0.80 | 0.82 | 0.83 | 0.79 | 0.81 | 0.81 | 0.83 |
| 5　　LShoulder | 0.54 | 0.60 | 0.79 | 0.80 | 0.53 | 0.63 | 0.80 | 0.80 | 0.52 | 0.54 | 0.77 | 0.78 |
| 6　　Lelbow | 0.39 | 0.78 | 0.81 | 0.83 | 0.66 | 0.78 | 0.79 | 0.81 | 0.41 | 0.72 | 0.79 | 0.79 |
| 7　　LWrist | 0.39 | 0.77 | 0.79 | 0.83 | 0.59 | 0.80 | 0.79 | 0.81 | 0.30 | 0.77 | 0.78 | 0.80 |
| Lower body | | | | | | | | | | | | |
| 9　　RHip | 0.42 | 0.48 | 0.64 | 0.64 | 0.38 | 0.51 | 0.66 | 0.64 | 0.40 | 0.49 | 0.59 | 0.64 |
| 10　　RKnee | 0.62 | 0.75 | 0.80 | 0.79 | 0.61 | 0.78 | 0.81 | 0.78 | 0.65 | 0.78 | 0.78 | 0.78 |
| 11　　RAnkle | 0.54 | 0.64 | 0.73 | 0.79 | 0.53 | 0.68 | 0.71 | 0.79 | 0.56 | 0.67 | 0.69 | 0.78 |
| 12　　LHip | 0.38 | 0.44 | 0.65 | 0.65 | 0.35 | 0.47 | 0.66 | 0.65 | 0.35 | 0.40 | 0.58 | 0.63 |
| 13　　LKnee | 0.50 | 0.72 | 0.80 | 0.79 | 0.41 | 0.71 | 0.81 | 0.79 | 0.51 | 0.66 | 0.79 | 0.78 |
| 14　　LAnkle | 0.43 | 0.54 | 0.71 | 0.78 | 0.40 | 0.65 | 0.71 | 0.78 | 0.47 | 0.55 | 0.68 | 0.78 |

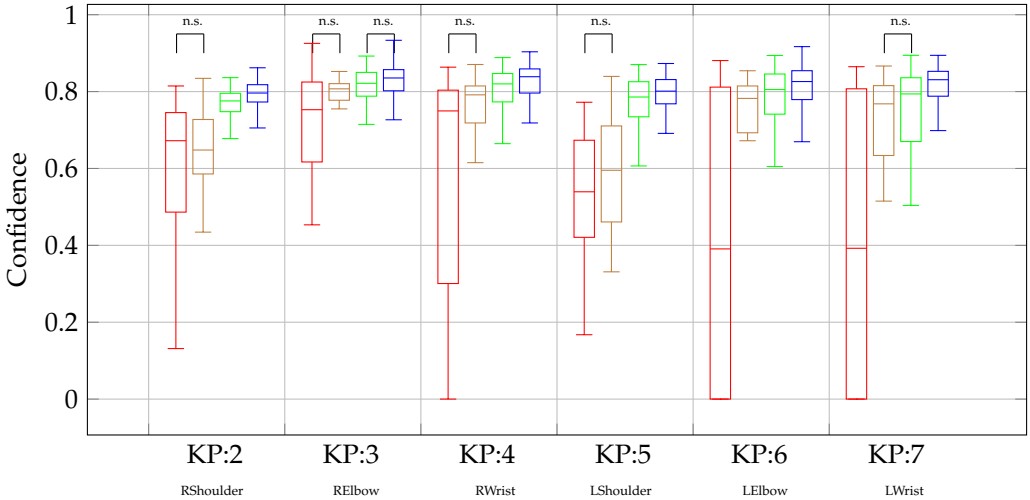

**Figure 4.** Confidence of detection of selected KP (keypoints)—front view, upper body. Camera views are shown in colour as follows: lying down ($N = 75$); on the knees ($N = 26$); sitting ($N = 186$); and standing ($N = 70$). All the other differences are statistically significant. Only differences that were not statistically significant (n.s.) are highlighted in the boxplots.

When interpreting the plots, it is important to note that the confidence correlates nonlinearly with the detection accuracy. If all keypoints in the image are clearly visible and accurately detected, confidence values in the range of 0.7–0.9 are commonly obtained. From a practical perspective of a single camera-based recording, the differences in accuracy associated with this range of confidences are not relevant. Even points with a confidence value above 0.5 can be considered correctly detected [9]. Lower confidence values are also associated with high-frequency keypoint jitter [36], but this effect can be easily filtered out because the body movements during rehabilitation exercises are slow relative to the sampling rate of the camera. False-positive detections or swapped keypoints can only be expected for low confidence values around 0.2 [34]. Small confidence values (0.1) are associated with guessed and occluded keypoints; the smaller the value, the more false positives detections are likely [9].

To summarize, as long as the value of the lower quartile is above 0.5, we can say that the combination of body position and camera view is practically usable.

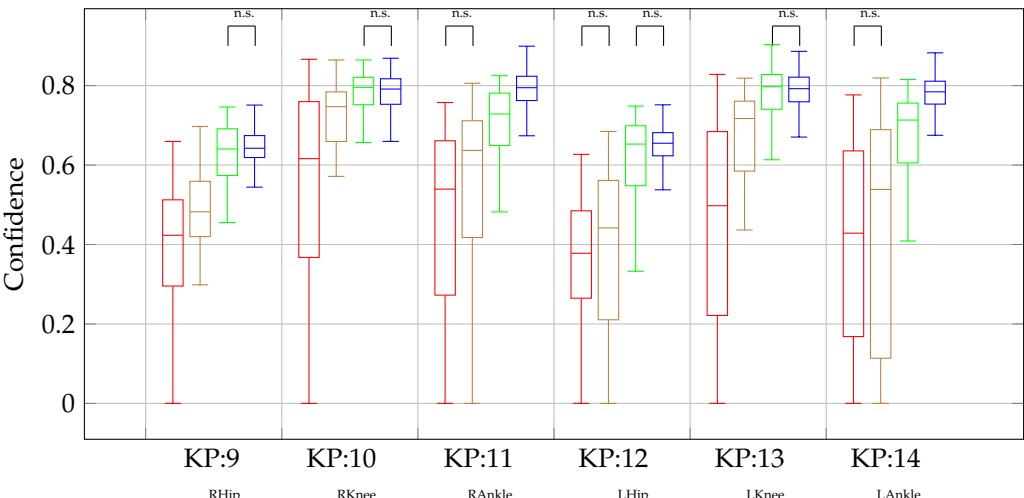

**Figure 5.** Confidence of detection of selected KP (keypoints)—front view, lower body. Camera views are shown in colour as follows: lying down (*N* = 75), on the knees (*N* = 26), sitting (*N* = 186) and standing (*N* = 70). All the other differences are statistically significant. Only differences that were not statistically significant (n.s.) are highlighted in the boxplots.

Interestingly we can see the differences between upper body points and lower body points. Upper body detection performs better for all camera views.

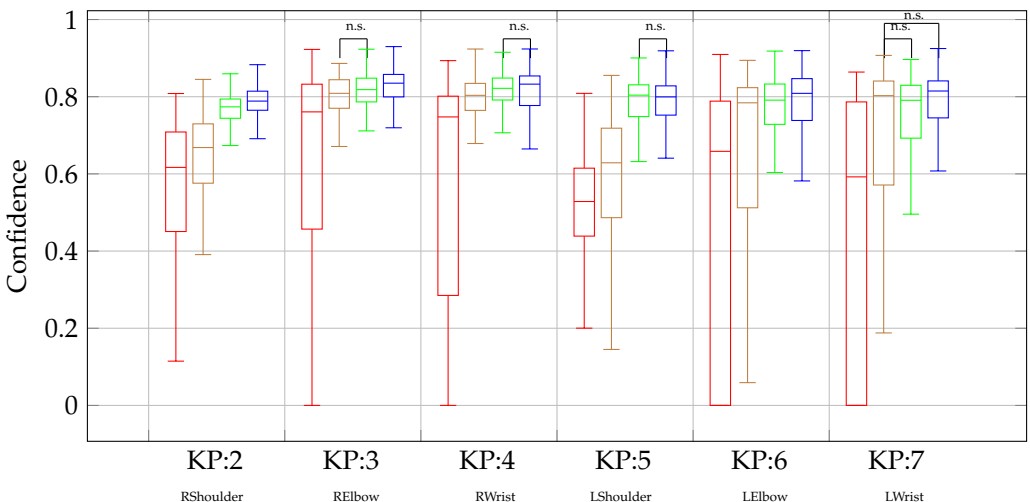

**Figure 6.** Confidence of detection of selected KP (keypoints)—¾ view, upper body. Camera views are shown in colour as follows: lying down (*N* = 177); on the knees (*N* = 95); sitting (*N* = 165); and standing (*N* = 490). All the other differences are statistically significant. Only differences that were not statistically significant (n.s.) are highlighted in the boxplots.

Non-significant differences in confidence are often found with standing and sitting positions, which is due to the fact that detection works very well.

If the value of the lower quartile of confidence is less than 0.5, it is likely that the detection will not work in all cases.

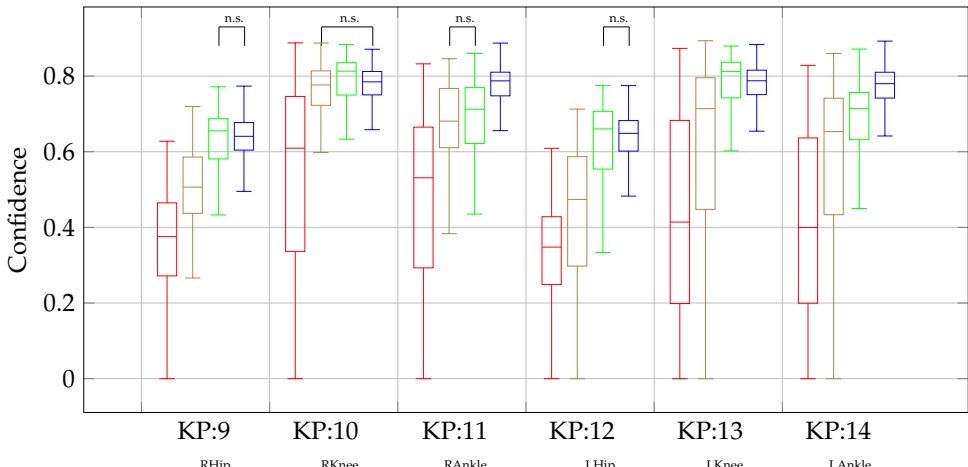

**Figure 7.** Confidence of detection of selected KP (keypoints)—¾ view, lower body. Camera views are shown in colour as follows: lying down (*N* = 177); on the knees (*N* = 95); sitting (*N* = 165); and standing (*N* = 490). All the other differences are statistically significant. Only differences that were not statistically significant (n.s.) are highlighted in the boxplots.

In all boxplots, we can observe an increasing tendency of confidence values between the groups of starting positions. The worst confidences are achieved by lying down (red), followed by on the knees (brown), and the best results are achieved by exercises performed in sitting (green) and standing (blue).

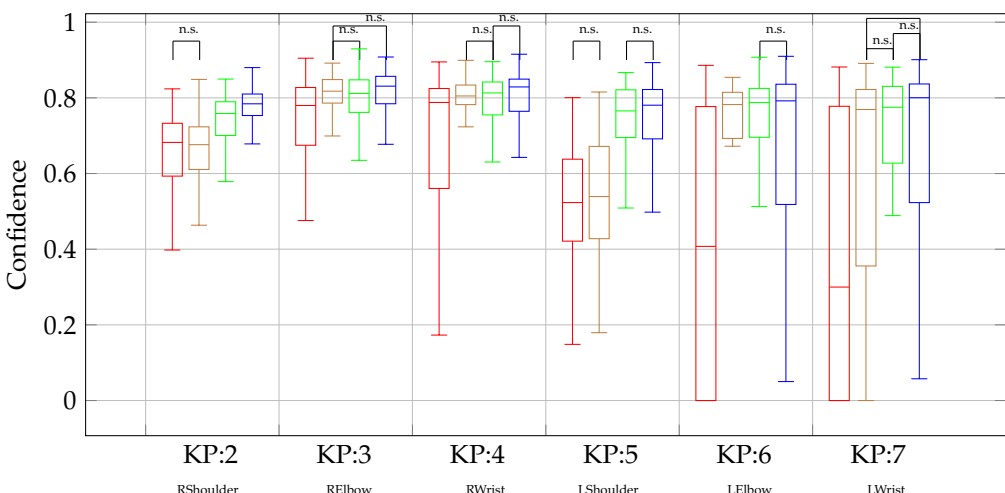

**Figure 8.** Confidence of detection of selected KP (keypoints)—side view, upper body. Camera views are shown in colour as follows: lying down (*N* = 215); on the knees (*N* = 92); sitting (*N* = 145); and standing (*N* = 297). All the other differences are statistically significant. Only differences that were not statistically significant (n.s.) are highlighted in the boxplots.

In the boxplots, the black line represents the groups that have no significant differences between them. These are mostly groups of exercises in sitting and standing positions. Thus, we can say that the detection is very reliable in both these groups and there are no significant differences between these groups. The achieved results clearly show that there are significant differences with respect to starting position.

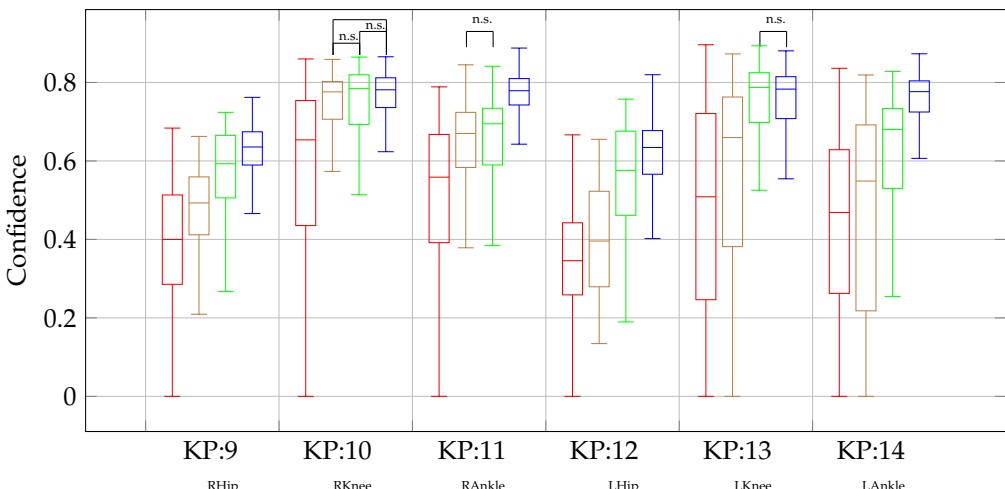

**Figure 9.** Confidence of detection of selected KP (keypoints)—side view, lower body. Camera views are shown in colour as follows: lying down (*N* = 215); on the knees (*N* = 92); sitting (*N* = 145); and standing (*N* = 297). All the other differences are statistically significant. Only differences that were not statistically significant (n.s.) are highlighted in the boxplots.

## 4. Discussion

The main objective of our research was to determine the usability of a camera-based system in a home environment. Since human position detection is captured by only one RGB camera, we were mainly interested in the influence of camera view and the position of the trainee, where we expected the greatest impact on the detection of keypoints. The quality of detection was determined by the confidence of detection of each keypoint.

Our main findings are the following: regardless of the camera view, the lying position comes out as the least detectable, followed by the position on the knees. The standing position is the most efficient, but the absolute differences against the sitting position are small. In the case of the camera view, the results were not so convincing. For the lying position and the position on the knees, the differences are not statistically significant in most cases, but no conclusions can be drawn because of the large variances.

For standing and sitting positions, the camera view from the side is a bit worse. From the data we have available, it is not possible to give a clear answer to the question of whether the confidences for the different camera views differ.

We also found out the difference in confidence between upper-body and lower-body keypoints. Confidence of lower-body keypoints is generally lower. This can be explained by the fact that the positions of the upper limbs are more variable than the position of the lower limbs. Generally speaking, joint positions are easier to establish if they are at an angle other than 180 degrees, which is typically the angle of the knee. The hips are not as visible as the shoulders and the ankles are often covered by shoes and trousers, while the position of the wrist can be very easily derived from the palm of the hand.

We can justify generalizing the results about views and postures given the high number of unique exercises, as opposed to works focusing on specific exercises, where only a few different types of exercises are involved.

Before applying the research results in practice, it is important to define several assumptions and limitations. They are closely linked to the application area of telerehabilitation in home settings. The first assumption and limitation at the same time is the use of a single simple camera (smart phone, tablet). The second assumption is the application use by nontechnical users that results in the requirement of simple control and setup of the application. These considerations led us to experiments analyzing the influence of the body position in relation to the camera and evaluation of many different exercises recorded by a single camera.

In the light of these facts, we are well aware of the limitations of the proposed approach, in particular the precise identification of certain motions in the front or side view. For example, abduction of the right arm cannot be well recognized in the side view from the left side, the range of straddling backward or angle of the knee cannot be precisely identified in frontal view.

For practical usability, it is important that there are not too many dropouts, i.e., that the joints are detected, and that they are not mistaken with another part of the body, e.g., the left and right limbs are swapped when viewed from the side, and so on. Another important aspect of the evaluation is that each individual exercise engages different parts of the body, thus only certain points are important for the analysis of the given exercise. The camera view is chosen so that the parts of the body being exercised are clearly visible while at the same time some parts of the body can be obscured. With the side camera view, the other side is often not visible.

Therefore, it can be assumed that points with a lower confidence value do not play a large role in the exercise. Just the fact that low confidences are found for individual joints does not necessarily mean that the exercise cannot be successfully evaluated. This is also the reason why we decided to present the results of individual joints and not evaluate the confidence of the whole exercise.

## 5. Conclusions

Despite the fact that there have been many recent publications describing the possibility of using a camera-based system for home rehabilitation, there has been no work to date that has validated the detection capability on a large dataset consisting specifically of videos of people performing rehabilitation exercises in front of a camera.

We validated the ability of the OpenPose algorithm to detect the keypoints of the human skeleton on more than two thousand videos of people performing rehabilitation exercises.

Based on our findings, we can say that OpenPose, for detection, is a sufficiently robust algorithm that is capable of detecting people during commonly performed exercises in a home environment. Only exercises performed in the lying and on-the-knees positions may not always be correctly detected. In this study, we also analyzed closely the basic landmarks of the human skeleton, see Table 3 and gave a summary of which keypoints are more reliably detectable. In that way, we provided an identification of the important points on the skeleton for each exercise, and, thus, offered a practical overview for designers of future camera-based telerehabilitation systems.

**Author Contributions:** Conceptualization, J.D. and J.A.; methodology, P.K.; software, J.A.; validation, J.A., L.L. and J.H.; formal analysis, P.K., J.H.; investigation, J.H.; resources, J.A.; data curation, J.A., J.H.; writing—original draft preparation, J.A., J.D.; writing—review and editing, L.L.; visualization, J.A.; supervision, L.L.; project administration, P.K.; funding acquisition, P.K. All authors have read and agreed to the published version of the manuscript.

**Funding:** This research was supported by a grant from the Ministry of Science & Technology, Israel and The Ministry of Education, Youth and Sports of the Czech Republic. The described research was supported by the project No. LTAIZ19008 Enhancing Robotic Physiotherapeutic Treatment using Machine Learning awarded in frame of the Czech–Israeli cooperative scientific research program (Inter-Excellence MEYS CR and MOST Israel).

**Institutional Review Board Statement:** Not applicable.

**Informed Consent Statement:** Not applicable.

**Data Availability Statement:** For research purposes, we only used body keypoints extraction, but did not store the videos themselves for future use. The videos are available on the Physiotools website [21]. Our statistical data are available along with our research.

**Conflicts of Interest:** The author declares no conflict of interest.

## Appendix A

**Table A1.** Camera view, *p*-values of comparison of starting positions. Ly—Lying; Kn—on the knees; St—standing; Si—sitting. The bold values show non-significant values.

| No. | Ly X Kn | Ly X Si | Ly X St | Si X Kn | St X Kn | St X Si |
|---|---|---|---|---|---|---|
| **Camera View FRONT** | | | | | | |
| Upper Body | | | | | | |
| 2 | **0.68** | <0.01 | <0.01 | <0.01 | <0.01 | <0.01 |
| 3 | **0.07** | <0.01 | <0.01 | <0.01 | <0.01 | **0.05** |
| 4 | **0.07** | <0.01 | <0.01 | <0.01 | <0.01 | <0.01 |
| 5 | **0.29** | <0.01 | <0.01 | <0.01 | <0.01 | <0.01 |
| 6 | <0.01 | <0.01 | <0.01 | <0.01 | <0.01 | <0.01 |
| 7 | <0.01 | <0.01 | <0.01 | **0.58** | <0.01 | <0.01 |
| Lower Body | | | | | | |
| 9 | <0.01 | <0.01 | <0.01 | <0.01 | <0.01 | **0.44** |
| 10 | <0.01 | <0.01 | <0.01 | <0.01 | <0.01 | **0.95** |
| 11 | **0.05** | <0.01 | <0.01 | <0.01 | <0.01 | <0.01 |
| 12 | **0.18** | <0.01 | <0.01 | <0.01 | <0.01 | **0.33** |
| 13 | <0.01 | <0.01 | <0.01 | <0.01 | <0.01 | **0.97** |
| 14 | **0.42** | <0.01 | <0.01 | <0.01 | <0.01 | <0.01 |
| **Camera View ¾** | | | | | | |
| Upper Body | | | | | | |
| 2 | <0.01 | <0.01 | <0.01 | <0.01 | <0.01 | <0.01 |
| 3 | <0.01 | <0.01 | <0.01 | **0.18** | <0.01 | <0.01 |
| 4 | <0.01 | <0.01 | <0.01 | <0.01 | <0.01 | **0.13** |
| 5 | <0.01 | <0.01 | <0.01 | <0.01 | <0.01 | **0.56** |
| 6 | <0.01 | <0.01 | <0.01 | <0.01 | <0.01 | <0.01 |
| 7 | <0.01 | <0.01 | <0.01 | **0.45** | **0.21** | <0.01 |
| Lower Body | | | | | | |
| 9 | <0.01 | <0.01 | <0.01 | <0.01 | <0.01 | **0.22** |
| 10 | <0.01 | <0.01 | <0.01 | <0.01 | **0.18** | <0.01 |
| 11 | <0.01 | <0.01 | <0.01 | **0.32** | <0.01 | <0.01 |
| 12 | <0.01 | <0.01 | <0.01 | <0.01 | <0.01 | **0.15** |
| 13 | <0.01 | <0.01 | <0.01 | <0.01 | <0.01 | <0.01 |
| 14 | <0.01 | <0.01 | <0.01 | <0.01 | <0.01 | <0.01 |

**Table A1.** *Cont.*

| No. | Ly X Kn | Ly X Si | Ly X St | Si X Kn | St X Kn | St X Si |
|---|---|---|---|---|---|---|
| **Camera View SIDE** | | | | | | |
| Upper Body | | | | | | |
| 2 | **0.96** | <0.01 | <0.01 | <0.01 | <0.01 | <0.01 |
| 3 | <0.01 | <0.01 | <0.01 | **0.38** | **0.13** | <0.01 |
| 4 | <0.01 | <0.01 | <0.01 | **1.00** | <0.01 | **0.07** |
| 5 | **0.16** | <0.01 | <0.01 | <0.01 | <0.01 | **0.37** |
| 6 | <0.01 | <0.01 | <0.01 | <0.01 | <0.01 | **0.51** |
| 7 | <0.01 | <0.01 | <0.01 | **0.29** | **0.07** | **0.36** |
| Lower Body | | | | | | |
| 9 | <0.01 | <0.01 | <0.01 | <0.01 | <0.01 | <0.01 |
| 10 | <0.01 | <0.01 | <0.01 | **0.46** | **0.23** | **0.85** |
| 11 | <0.01 | <0.01 | <0.01 | **0.22** | <0.01 | <0.01 |
| 12 | <0.01 | <0.01 | <0.01 | <0.01 | <0.01 | <0.01 |
| 13 | <0.01 | <0.01 | <0.01 | <0.01 | <0.01 | **0.54** |
| 14 | <0.01 | <0.01 | <0.01 | <0.01 | <0.01 | <0.01 |

**Table A2.** Starting position, *p*-values of comparison of starting positions. Fr—front view; Sid—side view; ¾—¾ view.

| | Lying | | | On the Knees | | |
|---|---|---|---|---|---|---|
| | Fr X ¾ | Fr X Sid | ¾ X Sid | Fr X ¾ | Fr X Sid | ¾ X Sid |
| Upper Body | | | | | | |
| 2 | **0.18** | **0.34** | <0.01 | **0.65** | **0.47** | **0.84** |
| 3 | **0.92** | **0.43** | **0.31** | **0.36** | **0.16** | **0.46** |
| 4 | **0.78** | <0.01 | <0.01 | **0.11** | <0.01 | **0.55** |
| 5 | **0.34** | **0.40** | **0.82** | **0.59** | **0.45** | <0.01 |
| 6 | **0.72** | **0.48** | **0.12** | **0.77** | **0.13** | <0.01 |
| 7 | 0.85 | 0.67 | 0.31 | 0.33 | 0.65 | 0.05 |
| Lower Body | | | | | | |
| 9 | **0.14** | **0.83** | **0.08** | **0.31** | **0.80** | **0.29** |
| 10 | **0.72** | **0.50** | **0.16** | **0.08** | **0.15** | **0.56** |
| 11 | **0.78** | **0.27** | **0.23** | <0.01 | **0.19** | **0.12** |
| 12 | **0.21** | **0.29** | **0.81** | **0.54** | **0.52** | <0.01 |
| 13 | **0.66** | **0.54** | **0.15** | **0.58** | **0.49** | **0.08** |
| 14 | **0.96** | **0.85** | **0.57** | <0.01 | **0.63** | <0.01 |
| | **Sitting** | | | **Standing** | | |
| | Fr X ¾ | Fr X Sid | ¾ X Sid | Fr X ¾ | Fr X Sid | ¾ X Sid |

**Table A2.** *Cont.*

|  | Lying | | | On the Knees | | |
|---|---|---|---|---|---|---|
|  | Fr X ¾ | Fr X Sid | ¾ X Sid | Fr X ¾ | Fr X Sid | ¾ X Sid |
| Upper Body | | | | | | |
| 2 | **0.57** | <0.01 | <0.01 | <0.01 | <0.01 | **0.07** |
| 3 | **0.69** | **0.16** | **0.15** | **0.81** | **0.16** | **0.14** |
| 4 | **0.74** | **0.23** | <0.01 | **0.12** | <0.01 | **0.14** |
| 5 | **0.09** | **0.18** | <0.01 | **0.35** | <0.01 | <0.01 |
| 6 | **0.29** | <0.01 | **0.18** | <0.01 | <0.01 | <0.01 |
| 7 | **0.93** | **0.49** | **0.38** | <0.01 | <0.01 | <0.01 |
| Lower Body | | | | | | |
| 9 | **0.50** | <0.01 | <0.01 | **0.42** | <0.01 | **0.09** |
| 10 | <0.01 | **0.19** | <0.01 | **0.11** | <0.01 | **0.17** |
| 11 | **0.28** | <0.01 | <0.01 | <0.01 | <0.01 | **0.12** |
| 12 | **0.51** | <0.01 | <0.01 | **0.08** | <0.01 | <0.01 |
| 13 | **0.12** | **0.40** | <0.01 | **0.11** | <0.01 | <0.01 |
| 14 | **0.92** | <0.01 | <0.01 | **0.29** | <0.01 | **0.08** |

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
