# Peer review of "Single Camera-Based Remote Physical Therapy: Verification on a Large Video Dataset"

_applsci, doi:10.3390/app12020799_

Round 1

Reviewer 1 Report

The topic is captivating and might interest readers, but it is metologically full of critical gaps

Abstract

1 Background statements in the abstract are missing

 2 Aim is cryptic. Reformulation is needed

2-5 This can be used as a background but the study objective is still unclear

  5 confidence is not an appropriate term. reliability?

Most important points of the human skeleton is non scientific. Anatomical landmarks? the importance on what basis? skeleton is biomechanically pretentious, better omit it

7-8 Methods missing. These are too strong conclusions, they need to be softened

10-14 Reformulate the section. I can understand the unstructured nature of the abstract but there is a need for clarity

Introduction

18-26 Brutal absence of bibliographic references

I suggest a single background section to be reformulated in its entirety. The scenario must be clarified, in which the softwares in the literature are present and well managed, but in the tele rehabilitative field, having examined the pose of subjects who often have pathologies, it was necessary to conduct this study.

42-43 Coyler et al. described … 

Methods

78-80 Introduction rationale

The study design is missing, where it was conducted, how the data were collected and processed

98-99 Why conjugate the verb in the future? But above all because 12, there is a need for bibliographic references

123 Do you boast a very large database and then use non-parametric tests? Then why is the reliability of this analysis protocol not mentioned?

Results 

unfortunately disconnected, how big is this video database?

Discussion

174 By convention, the discussion does not begin with the limitations, but with the goal and the major findings of the results .. after the discussion of this.. the limitations to be introduced before the conclusions ..

Author Response

Thank you for your meaningful comments.

Abstract

1 Background statements in the abstract are missing

 2 Aim is cryptic. Reformulation is needed

2-5 This can be used as a background but the study objective is still unclear

  5 confidence is not an appropriate term. reliability?

Most important points of the human skeleton is non scientific. Anatomical landmarks? the importance on what basis? skeleton is biomechanically pretentious, better omit it

7-8 Methods missing. These are too strong conclusions, they need to be softened

10-14 Reformulate the section. I can understand the unstructured nature of the abstract but there is a need for clarity

All of the above suggestions have been incorporated and the abstract has been completely revised.

Introduction

18-26 Brutal absence of bibliographic references

I suggest a single background section to be reformulated in its entirety. The scenario must be clarified, in which the softwares in the literature are present and well managed, but in the tele rehabilitative field, having examined the pose of subjects who often have pathologies, it was necessary to conduct this study.

42-43 Coyler et al. described … 

The introduction has been significantly revised. It was mainly to add citations and to elaborate the motivation for the research.

Methods

78-80 Introduction rationale

The study design is missing, where it was conducted, how the data were collected and processed

We have added a paragraph explaining the rationale for our research and added a diagram describing the study design and the steps in a simplified manner.

98-99 Why conjugate the verb in the future? But above all because 12, there is a need for bibliographic references

The English grammar has been corrected and a reference has been added both to the table describing the 12 points and to the literature from which we determined the points

123 Do you boast a very large database and then use non-parametric tests? Then why is the reliability of this analysis protocol not mentioned?

We added an explanation that the nonparametric test was used because of the heterogeneous sizes in the groups and the non-normal distribution of the data.

Results 

unfortunately disconnected, how big is this video database?

An explanatory paragraph explaining what the results represent was added at the beginning of the section. All figures with boxplots were supplemented with the values of the videos in the group. This improved the readability of the results.

Discussion

174 By convention, the discussion does not begin with the limitations, but with the goal and the major findings of the results .. after the discussion of this.. the limitations to be introduced before the conclusions ..

The comments have been modified in accordance with the above-mentioned proposal.

Reviewer 2 Report

The authors present an experimental study to demonstrate the efficacy of detecting specific points for remote assessment for rehabilitation schemes.

  1. The introduction section must be enhanced to include why is important to study software and hardware tools for performing telerehabilitation.
  2. Modern rgb cameras have more than 6 megapixels. It is expected that the database include videos taken with cameras that have the aforementioned feature.
  3. How can the low confidence values be interpreted? Does this value indicate that the position cannot be properly processed using the software?
  4. How do the authors propose to evaluate the efficacy of the rehabilitation schemes now that they prove that for certain body positions is possible to detect them?

Author Response

Thank you for your relevant comments.

  â–² Thank you for your relevant comments.

  1. The introduction section must be enhanced to include why is

important to study software and hardware tools for performing

telerehabilitation.

The introduction section has been completely redesigned and supplemented with additional references. Motivation for why this study is important to the field of telerehabilitation has been added.

  1. Modern rgb cameras have more than 6 megapixels. It is

expected that the database include videos taken with cameras

that have the aforementioned feature.

Video of this quality is sufficient because the OpenPose system reduces the resolution of the video before processing by the neural network. We also add a text:

Our aim is to evaluate the practical usability of recordings made in an uncontrolled home environment so we use no additional constraints on video quality.

  1. How can the low confidence values be interpreted? Does this

value indicate that the position cannot be properly processed

using the software?

Yes, low confidence values indicate that the keypoint position can not be properly detected. We have added a reference to work [33], that shows how does the confidence value to keypoint detection precision.  

  1. How do the authors propose to evaluate the efficacy of the

rehabilitation schemes now that they prove that for certain body

positions is possible to detect them?

The aim of this paper was not to evaluate the effectiveness of rehabilitation schemes but to investigate whether detection is possible. Possible directions to proceed are based on the summarized results. That is, primarily that OpenPose is robust enough to detect key points in the home environment.

Reviewer 3 Report

The paper is interesting as it demonstrate the usability of markerless pose recognition for something more substantial than entertainment. However this a survey only, comparing alrogithms for a predefined use in rehabilitation. The paper is well written and may be of interest as a survey. I have only small remark - make one more proof-reading.

Author Response

The paper is interesting as it demonstrate the usability of markerless pose recognition for something more substantial than entertainment. However this a survey only, comparing alrogithms for a predefined use in rehabilitation. The paper is well written and may be of interest as a survey. I have only small remark - make one more proof-reading.

Thank you for your kind comment. We've made a lot of adjustments to the paper to make the text easier to read, correcting the English and reformulating some incomprehensible passages.

Round 2

Reviewer 1 Report

Thank you for the important effort in the first revision, but before suggesting it suitable for publication I wanted to re-emphasize the weight of the reliability assessment (intra-rater and inter-rater), just when you talk about reliability

Abstract

Lines 1-2 Remove first sentence.

Lines 8-9 Anatomical landmarks

Line 16 Might also indicate

Introduction

As previously pointed out, do not leave important statements without a bibliographic reference, because it conveys little solidity to the sentence itself.

For example line 23 Line, I suggest expanding the statement with a ref: “The general concept of remote rehabilitation using motion capture (MoCap) systems has undergone a turbulent change in recent years, as there are several tools for three-dimensional assessments, including sophisticated automation technologies and algorithms, often costing time, expensive equipment and inapplicable inconvenience to the daily practice.” (ref: https://doi.org/10.1016/j.jelekin.2020.102485 )

Line 34 ref: https://doi.org/10.3390/s16020208

Line 47 repeat ref [11]

Line 52 “physiotherapists need to monitor and modulate the rehabilitation intervention”
ref: http://dx.doi.org/10.1108/JET-11-2020-0047

Paragraph 60-87 the title must be removed because it is not influential for the manuscript and the lines must be moved before line 43. The reason is that the background (the whole overview on the instruments, must be placed before the rationale and the study objective (which are at line 52-53 ))

Methods

Line 88 Title of the paragraph is “Design”.. remove video database.

Line 100 [19] is not a reference, put “PhysioTools©, Product ID RG-PT1ENG, General Exercises Second Edition (English), Tampere, 339 Finland” put the sentence in round brackets after quoting and recalling the name just like a software .. for example R 3.62 (R fundation, Vienna, Austria)

Line 112 I suggest “Video Analysis”

150 List and describe all software used, including OpenPose, for R tests? STATA?

159 it is not necessary to put results in the methods, but the methodological steps carried out must be described slavishly

A question that occurs to me is if you have a large sample, why are non-parametric tests used?

Results

At the beginning I would recommend an objective overview of the videos analyzed (how many, all the 1168 videos?).

171-172 whether it is large or not, or any other comment on the results, let the reader ascertain.

The last important perplexity is on the real reliability of the study, so I ask you why not evaluate the intra-rater and inter-rater reliability of the analysis. He remains a gamechanger .. please explain

Author Response

Thank you for your feedback. Please see our comment below. 

Abstract

Lines 1-2 Remove the first sentence.

We have removed the first sentence

Lines 8-9 Anatomical landmarks

We have replaced “appendicular skeleton landmarks” with “anatomical landmarks”

Line 16 Might also indicate

We have updated the sentence

Introduction

As previously pointed out, do not leave important statements without a bibliographic reference, because it conveys little solidity to the sentence itself.

For example line 23 Line, I suggest expanding the statement with a ref: “The general concept of remote rehabilitation using motion capture (MoCap) systems has undergone a turbulent change in recent years, as there are several tools for three-dimensional assessments, including sophisticated automation technologies and algorithms, often costing time, expensive equipment and inapplicable inconvenience to the daily practice.” (ref: https://doi.org/10.1016/j.jelekin.2020.102485 )

Thanks for the suggestion, we have expanded by suggested text and ref.

Line 34 ref: https://doi.org/10.3390/s16020208

Thanks for the suggestion, added as suggested.

Line 47 repeat ref [11]

We have repeated the ref. 

Line 52 “physiotherapists need to monitor and modulate the rehabilitation intervention”

ref: http://dx.doi.org/10.1108/JET-11-2020-0047

We have extended by suggested ref.

Paragraph 60-87 the title must be removed because it is not influential for the manuscript and the lines must be moved before line 43. The reason is that the background (the whole overview on the instruments, must be placed before the rationale and the study objective (which are at line 52-53 ))

The paragraph has been moved to the introduction section.

Methods

Line 88 Title of the paragraph is “Design”.. remove video database.

We have updated the title

Line 100 [19] is not a reference, put “PhysioTools©, Product ID RG-PT1ENG, General Exercises Second Edition (English), Tampere, 339 Finland” put the sentence in round brackets after quoting and recalling the name just like a software .. for example R 3.62 (R fundation, Vienna, Austria)

We have put it directly into the text in rounded quotes.

Line 112 I suggest “Video Analysis”

We have changed the title.

150 List and describe all software used, including OpenPose, for R tests? STATA?

It is clear from the text that we have used OpenPose for keypoint extraction. We have added reference to MATLAB because we used it in all our calculations.

159 it is not necessary to put results in the methods, but the methodological steps carried out must be described slavishly

A question that occurs to me is if you have a large sample, why are non-parametric t 

We use the non-parametric test because the data in groups do not have a normal distribution. We have used the Shapiro–Wilks test to determine the normality. We have mentioned it explicitly in the text. 

There has been some discussion going on in regard to the use of non-parametric tests and a large sample size. 

While large sample size and central limit theorem guarantee the normal distribution of the sample mean values it does NOT guarantee the normal distribution of samples in the population. 

Therefore, even if the sample size is sufficient, it is still recommended that the results of the normality test be checked first https://www.ncbi.nlm.nih.gov/pmc/articles/PMC6676026/

Results

At the beginning I would recommend an objective overview of the videos analyzed (how many, all the 1168 videos?).

We have added the following text: In our study, we analyzed a total of 2133 videos. Each video shows only one trainee performing a unique exercise. Each video belonged to one of the "Camera View" and one of the "Starting position" groups.

171-172 whether it is large or not, or any other comment on the results, let the reader ascertain.

The last important perplexity is on the real reliability of the study, so I ask you why not evaluate the intra-rater and inter-rater reliability of the analysis. He remains a gamechanger .. please explai

Thank you for the important effort in the first revision, but before suggesting it suitable for publication I wanted to re-emphasize the weight of the reliability assessment (intra-rater and inter-rater), just when you talk about reliability

OpenPose itself has almost perfect test-retest reliability within the device. We have added a small paragraph mentioning this. 

Categorization of videos was done by a single rater and border cases that didn't seem to belong to either group were excluded. We updated the sentence in the manuscript.

It would have been methodologically better to have multiple raters do the categorization into groups. We initially had more raters, but everyone rated the selected samples equally, so we decided that it would be redundant and time-consuming work to use multiple raters.

The categorization itself is not an expert evaluation -  the category is obvious at first glance at the video. We felt it is more important to not have any videos that would be in the middle or outside of the categories. But since we defined the categories by looking at the videos, only a small portion of the videos was excluded, as the selected categories were typical for the majority of the videos.  

Reviewer 2 Report

The authors have addressed all the concerns expressed.

Author Response

Thanks for your feedback.

Round 3

Reviewer 1 Report

Dear authors,
I can congratulate you on the effort in the review. The manuscript is more solid, I can only suggest that:
- add explanation of abbreviations to all figures to ensure complete understanding (KP, although it is explained in the text)
- underline the fact that OpenPose remains one of the few opensource and openaccess

Best regards

Author Response

Thank you very much for your precise reviews. Your suggestions have really helped to improve the quality of the article. 

We have also incorporated the last suggestions.

Please see the last version.

Best regards